# On the Track of New Endoscopic Alternatives for the Treatment of Selected Gastric GISTs—A Pilot Study

**DOI:** 10.3390/medicina57060625

**Published:** 2021-06-16

**Authors:** Artur Raiter, Katarzyna M. Pawlak, Katarzyna Kozłowska-Petriczko, Jan Petriczko, Joanna Szełemej, Anna Wiechowska-Kozłowska

**Affiliations:** 1Department of Endoscopy, Specialist Hospital of Alfred Sokolowski, 58-309 Wałbrzych, Poland; artur.raiter@outlook.com (A.R.); joannaszelemej@gmail.com (J.S.); 2Endoscopy Unit, Department of Gastroenterology, Ministry of Interior and Administration, ul. Jagiellońska 44, 70-382 Szczecin, Poland; annamwk@wp.pl; 3Department of Gastroenterology and Internal Medicine, SPWSZ Hospital, 70-382 Szczecin, Poland; kasia-petriczko@outlook.com; 4Department of Plastic, Endocrine and General Surgery, Pomeranian Medical University, 70-382 Szczecin, Poland; jan.petriczko@gmail.com

**Keywords:** gastric GIST, endoscopic resection, ESD, endoscopic suturing, endoscopic ultrasound

## Abstract

*Background and Objectives*: GISTs (Gastrointestinal stromal tumors) are the most common mesenchymal gastrointestinal tract tumours and are mainly located in the stomach. Their malignant potential depends on size, location, and type. Endoscopic techniques are a less invasive modality for patients not eligible for surgery. ESD (endoscopic submucosal dissection) is mainly used for the removal of smaller GISTs, with intraluminal growth and a more superficial location. Thus, R0 resection capability in some cases may be not sufficient, limited by tumour size, location in the gastric wall, and its connection level with the muscularis propria. In such cases, an endoscopic full-thickness resection can become a new alternative. In this retrospective pilot study, we evaluated ESD and hybrid resection techniques in terms of safety, efficacy, and disease recurrence for selected types of gastric GISTs. *Materials and Methods:* A retrospective comparison was conducted in a group of patients who underwent ESD or a hybrid technique combining endoscopic resection with endoscopic suturing using the OverStitch system (HT) for type II or III gastric GISTs. A total of 21 patients aged 70 ± 8 years underwent endoscopic resection. Seventeen lesions were treated with ESD and four with the HT. *Results:* R0 resection was achieved in all patients treated using HT (type III lesions) and in 53% of those treated with ESD (*p* = 0.08). None of the type III lesions treated with ESD were excised with R0. Lesions treated with R0 ESD resections were significantly smaller (1.76 ± 0.35 cm) than those with R1 ESD resections (2.39 ± 0.40 cm) (*p* < 0.01). The mean lesion size treated with the HT was 2.88 ± 0.85 cm. *Conclusions:* HT may be a new resection modality for large gastric GISTs with high muscularis propria connection grades. Further studies are required to evaluate its safety and efficacy and to form precise inclusion criteria for endoscopic resection techniques.

## 1. Introduction

Gastrointestinal stromal tumours (GISTs) are the most common mesenchymal lesions of the gastrointestinal (GI) tract, constituting nearly 80% of mesenchymal pathologies [1]. Due to possible malignant transformation [2] and unexpected, rapid growth observed in approximately 3.7% of GISTs, close surveillance and early excision when necessary are the mainstay of management [3].

According to the latest recommendations of the European Society for Medical Oncology, the management of GISTs should include wide excision with confirmed tumour-free margins (R0) as the principal treatment goal for resectable tumours, as it minimises the recurrence rate [4,5,6]. Intra-procedural complications, such as tumour ruptures and perforations, also negatively affect prognosis [6]. While surgical laparoscopic wedge resection remains the gold standard for GIST treatment, in case of non-operable lesions of varying circumstance, endoscopic resection, such as endoscopic submucosal dissection (ESD) modalities, is minimally invasive and allows for R0 resection with no complications [7,8].

Factors influencing the choice of resection technique include lesion size, lesion type, and location [5]. The National Comprehensive Cancer Network Guidelines suggest resection of all tumours over 2 cm in diameter or in the case of malignancy suspicion and/or progression during the follow-up period, regardless of the tumour size [9]. For endoscopic resection, it is crucial to evaluate the tumour’s relationship to the muscularis propria (MP) and the type of growth according to the Kim classification using endoscopic ultrasound (EUS) [10]. When the connection to the MP is thin and the growth is intraluminal (type I), a safe and effective resection can be performed using a variety of techniques, ranging from endoscopic mucosal resection to ESD [10]. Type II and III GISTs have more profound intramuscular growth and wider connection with the MP, increasing the risk of incomplete resection and perforation during endoscopic resection (Figure 1) [7,10]. Deep resection during ESD, which is necessary to provide adequate excision, is associated with a perforation rate of up to 12% [11,12,13,14]. Thus, tumours infiltrating the deeper layers of the stomach wall may limit the clinical usefulness of some techniques [15]. In this study, we retrospectively compared ESD to the endoscopic hybrid resection technique in terms of safety, efficacy, and disease recurrence for selected types of GISTs.

## 2. Materials and Methods

This is single-centre, open-label, retrospective case study of consecutive patients who underwent ESD or the endoscopic hybrid resection technique of gastric GISTs between 2017 and 2019. The selection of patients was based on the evaluation of particular clinical features and optimal care of patients after multidisciplinary committee presentation and qualification. All procedures were performed at two endoscopic units (Hospital of the Ministry of Interior and Administration Szczecin, Poland and Specialist Hospital of Alfred Sokolowski, Wałbrzych, Poland). By decision of the local Institutional Review Board in Szczecin (IRB KB/0012/78/11/2020/Z), no formal approval was needed. The study was developed using the STROBE guidelines [16] (Appendix A), in accordance with the Declaration of Helsinki. Informed consent was obtained from all patients prior to any procedures.

### 2.1. Study Group Characteristics

All GISTs were evaluated with EUS and confirmed by EUS-guided biopsy. Systemic staging was based on computed tomography to exclude advanced disease. We include all type II and III GISTs, resected by ESD and hybrid techniques. Patients were retrospectively divided into two groups. Group I comprised all patients who had undergone ESD. Prior to the introduction of the hybrid technique, patients with type III lesions not eligible for surgery were also treated with ESD, thus being included into the first group. After the introduction of the hybrid approach, such patients were offered the option of resection with endoluminal suturing and comprised group II.

Twenty-one patients with gastric type II or III GISTs, with a diameter of > 12 and < 40 mm and confirmed local disease without metastases or infiltration of surrounding tissues in imaging were included in the study. Patient characteristics and tumour data are presented in Table 1.

### 2.2. Resection Techniques

All procedures were performed by the same expert endoscopist (A.R.), with prior experience of ESD for upper and lower GI mucosal lesions. Patients were under general anaesthesia while in a supine position. For the ESD, a GIF-HQ190 gastroscope (Olympus Medical Systems, Hamburg, Germany), DualKnife (Olympus), and Coagrasper (Olympus) were used. For the hybrid resection technique, an additional GIF-2TH180 gastroscope (Olympus) and an OverStitch system (Apollo, TX, USA) were used.

All the patients received a standard pre-operative dose of second-generation cephalosporins. ESD was performed according to commonly accepted standards with a clip closure of the defects (Figure 2). The hybrid technique combined elements of ESD with endoscopic endoluminal suturing. The hybrid technique began by marking tumour borders (using electrocautery), followed by an indigo carmine solution injection into the submucosal layer. A standard ESD procedure was continued until the connection with the MP layer was revealed. Next, the standard endoscope (GIF-HQ190) was replaced with the double-channel endoscope (GIF-2TH180) with OverStitch installed. Gastric wall duplication (i.e., doubling the layers of the MP and serosa) was achieved through continuous suture below the tumour (Figure 3a). The muscle layer was then dissected between the lesion and the sutures (Figure 3b). The specimen was removed from the stomach in a single piece through the oesophagus using a standard endoscope and a Roth net (US Endoscopy, STERIS, Mentor, Ohio, USA). The largest lesions were partially secured within the Roth net and also removed in one piece.

R0 was defined as a complete tumour excision, free of residual tumour cells on both the horizontal and vertical margins, confirmed by a qualified pathologist. Additionally, for tumours resected with the ESD technique, R0 was assessed by biopsies of the resection bed. Regarding the hybrid technique, the wide margins achieved during resection did not necessitate additional tissue sampling.

### 2.3. Outcomes

The primary outcome was a comparison of the efficacy of ESD and hybrid gastric GIST resection in terms of achieving R0. The secondary outcome was the assessment of the effect of lesion size on the rate of R0 resection.

### 2.4. Statistical Analysis

The Kolmogorov–Smirnov test was used to observe the distribution of variables. Continuous variables are expressed as means ± standard deviations (SDs). Parametric tests (Student’s *t*-tests and ANOVAs) were used for the assessment of differences between numerical variables with normal distributions, and non-parametric tests (Mann–Whitney or Kruskal–Wallis tests) were used for variables with non-normal distributions. Categorical variables are presented as values and percentages. To evaluate the significance of the associations between categorical variables, Fisher’s exact test and Pearson’s χ^2^ test were used. The Spearman correlation test (ρS) was used to assess the relationship between two variables. All *p*-values are two sided, and *p*  <  0.05 was considered significant. All statistical analyses were performed using STATA 11 (StataCorp., College Station, TX, USA).

## 3. Results

### 3.1. Characteristics of the Study Population

A total of 21 patients were included in the analysis. Patient characteristics are shown in Table 1. The average age was 70 ± 8 years, with 11 tumours (52%) diagnosed in male patients. Seventeen lesions were treated using the ESD method and four with the hybrid technique. The mean lesion size in the ESD group was significantly smaller (2.05 ± 0.49 cm) in comparison to that of those treated using the OverStitch system (2.88 ± 0.85 cm). Of the 17 lesions treated with ESD, 11 (65%) were located in the body of the stomach, with the rest in the antrum. In the group treated with hybrid technique, one lesion was located in the stomach body, one in the antrum, and two in the fundus. All lesions treated with the OverStitch system were type III. In comparison, only two type III lesions (12%) were excised using ESD (*p* < 0.001). No intra-operative or delayed complications were observed in either group.

### 3.2. Comparison between ESD and the Hybrid Technique

R0 resection, i.e., therapeutic success, was achieved in all the patients treated with the OverStitch system and in 53% (9 of 17) of the patients treated with ESD (Table 1). All lesions treated successfully with ESD (R0 resection) were type II in comparison to lesions treated with the OverStitch, which were all type III (*p* < 0.005) (Figure 4). Furthermore, lesion size in patients treated with ESD with R0 resection (1.76 ± 0.35 cm) differed significantly (*p* < 0.005) from the lesion size in patients with R1 ESD resection (2.39 ± 0.40 cm) (Figure 5). The average ESD procedure time was 86.2 ± 33.9 min, which was shorter than 122.5 ± 78.5 min for the hybrid technique, but these results did not achieve statistical significance. However, ESD procedures that did not achieve R0 resection lasted significantly longer (109 ± 31 min) than those with therapeutic success (66 ± 21 min) (*p* < 0.005). The correlations between procedure time and lesion size in different procedure types are presented in Figure 6. A larger lesion size prolonged the procedure with the highest correlation rate for the OverStitch procedure (R = 0.95) followed by ESD R0 resection (R = 0.7). In patients with R1 resection, a secondary EFTR of the scar was performed with therapeutic success. All patients presented as disease free at a 12-month follow-up exam. No intra-operative or delayed complications in either group were observed.

## 4. Discussion

We found that therapeutic success during the initial procedure was achieved in all the tumours treated with HT and in half of the patients treated with ESD. A larger tumour size and a wider connection to the MP (type III tumours) were identified as factors associated with a failure of ESD R0 resections.

The advancements in minimally invasive endoscopic treatment techniques allow for the safe removal of significantly larger GI tumours than before. ESD is a resection method for early tumorous lesions limited to the superficial layers of the intestinal lining. Previously, we reported [11] that a precise pre-procedural evaluation of a tumour/MP connection using EUS was crucial for achieving complete resection. ESD enables an R0 resection of large sub-mucosal tumours, provided that they have a relatively narrow connection with the MP, such as type I tumours. In the case of lesions with a broader connection with the MP (type II and III), it is difficult to achieve a safe R0 resection [11]. A similar correlation was observed by An et al. [12]. Furthermore, recent studies have shown an inverse correlation between the efficacy of endoscopic resection and the diameter of the tumour. Lee et al. [13] reported that a complete endoscopic resection of GISTs with ESD was not achievable with a mean tumour size of 27.5 mm. Białek et al. [14] assessed ESD in terms of the efficacy of treatment. In tumours connected to the MP, the achieved R0 resection rate was 68.2%, which decreased as lesion type and tumour size increased. Successful R0 resections were predicted by observing no or narrow tumour connections with the underlying MP during EUS (OR = 35.0, 95% CI: 3.7–334.4, *p* = 0.001).

The R0 resection was not achieved in 40% of the type II and in all of the type III tumours within the ESD group. This indicates that in type II lesions that could be resected using standard endoscopic methods, ESD does not guarantee a clean resection margin. This may be due to having only a small margin of tissue removed with the tumour during ESDs. GISTs do not have a well-defined capsule; therefore, in cases of R1 resection, it is necessary to perform scar tissue removal, for example, by using (EFTR) with a dedicated Ovesco^®^ set (Tübingen, Germany). By comparison, Shichijo et al. [15] presented the results of an alternative hybrid variant of the EFTR technique using elements of ESD, clipping, and ligating with endoloop. In 62% of the tumours, the resection margins were indeterminate, even in cases of deep resections followed by perforation.

We also found that the size of the tumours successfully treated with ESD (R0) was significantly smaller than in the ESD-R1 group. On the other hand, the mean tumour size treated with the Apollo OverStitch system (R0 resections) was significantly larger, up to 40 mm. We speculate that, when using HT, even larger lesions may be amenable to R0 endoscopic resection. In our study, we limited the size of resection lesions to 40 mm in order to be able to remove them through the oesophagus as one piece.

Several studies evaluating ESD efficacy have estimated recurrence rates up to 6.7% [14,16,17,18]. Furthermore, the long-term efficacy of GIST removal by ESD is controversial, especially in cases of gastric wall defects and tumour ruptures [19]. The most common complications include perforations, which occur in 1.2–9.7% of cases, and bleeding (up to 15% cases) [12,14,15,17,18,20,21,22,23].

In our study, we observed neither early nor delayed complications in either of the groups. Therefore, the hybrid technique was chosen for the removal of the largest and most difficult lesions with wider MP connections. Duplication of the stomach wall under the tumour is essential for providing a safe plane for dissection between the duplicated stomach wall and the tumour, thereby effectively preventing perforation and bleeding. Moreover, the technique ensures a safe excision of the lesion with an appropriately wide margin of the surrounding tissue. This increases the likelihood of an R0 pathology evaluation and reduces the recurrence rate, as observed in our follow-up.

The hybrid approach does have some limitations, including increased cost and longer surgical times. An average ESD procedure was significantly quicker to complete (86.2 ± 33.9 vs. 122.5 ± 78.5 min), which was due to larger lesion sizes, wider MP connections, and the need to change endoscopes during the hybrid procedure. The hybrid technique costs nine times more than a standard ESD and requires a highly skilled endoscopist; however, in comparison, the surgical approach is invasive, even more time consuming, and expensive. Moreover, the therapeutic success rates of hybrid endoscopic techniques are comparable to those of standard surgical treatments [24]. Preventing perforations during hybrid resections also contributes to shorter post-procedure hospital stays [23,25,26], especially in comparison with other previously reported multi-device EFTR techniques, which create controlled perforations [23,26].

There are limitations of this study. First, retrospective studies have the inherent potential for bias and data incompleteness. However, as we focus on advanced resection methods, we keep a carefully collected patient database and perform dedicated follow-ups of all our patients. Second, the sample size is relatively small. This is related to the included tumour types (II and III) and the infrequent occurrence of such lesions. Initially, such patients were qualified for surgical excisions, but as the hybrid method is novel and not recommended as a first-line treatment, patients included in our study were those who had been disqualified from a surgical approach. Nonetheless, our results show that ESD does not result in R0 in up to 40% of cases as shown in other studies. More importantly, the large success rate of R0 in larger lesions supports the notion that proceeding primarily with this technique in most lesions would result in higher R0 rates and avoid secondary HT to complete resection in cases of R1.

## 5. Conclusions

The hybrid technique combining endoscopic resection and endoluminal suturing appears to be a potential alternative for gastric GISTs with a large size and high MP connection grades (type > I), with advantages over ESD. Further studies are needed to evaluate the safety and efficacy of the procedure and to form precise inclusion criteria for the hybrid procedure as an alternative treatment to surgical resection.

## 6. What Is Known

-Gastrointestinal stromal tumours (GISTs) are the most common mesenchymal lesions of the gastrointestinal tract, constituting nearly 80% of mesenchymal pathologies.-Due to possible malignant transformation and unexpected, rapid growth, close surveillance and early excision are the mainstay of management.-The less invasive procedure includes various endoscopic resection techniques, such as ESD, EFTR, and STER.

## 7. What We Found

-The hybrid technique combining endoscopic resection and endoluminal suturing appears to be a potential alternative for selected gastric GISTs with a large size and a high MP connection grade (type > I), with advantages over ESD.

## Figures and Tables

**Figure 1 medicina-57-00625-f001:**
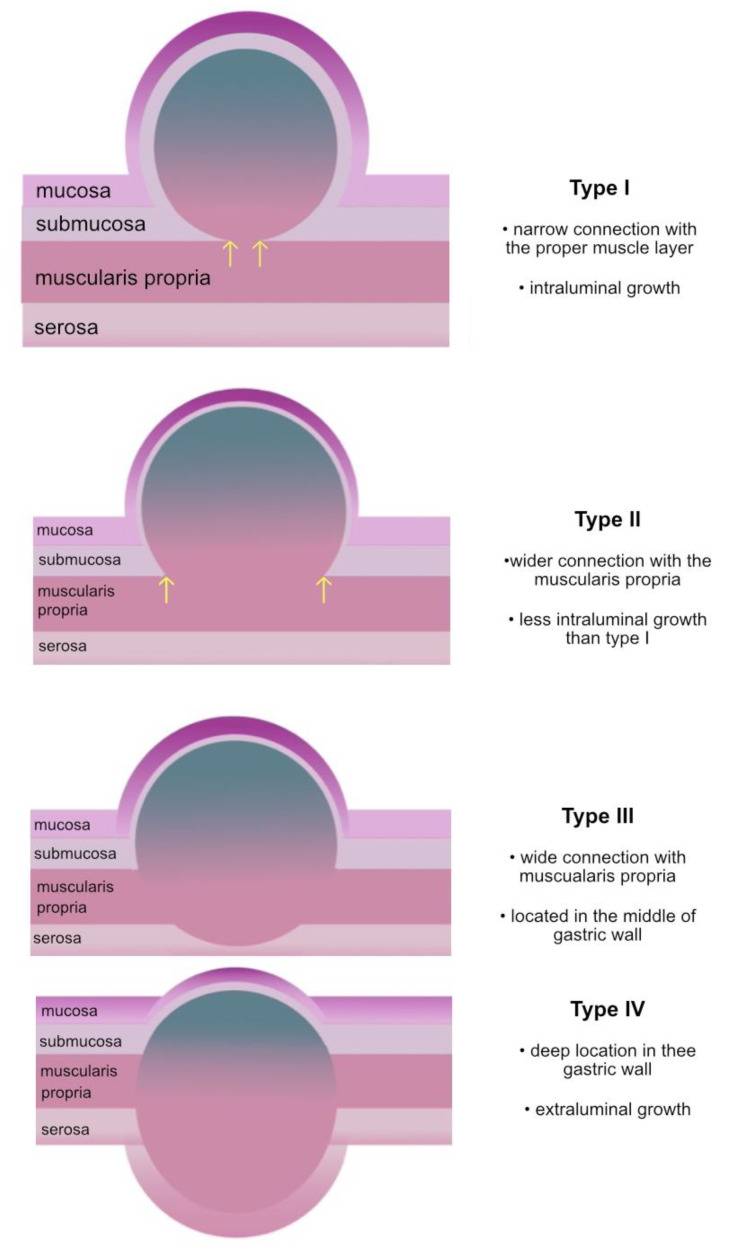
Type of GISTs according to the location in stomach wall (adapted from Kim [10], with permission from Baishideng Publishing Group Inc., 2021).

**Figure 2 medicina-57-00625-f002:**
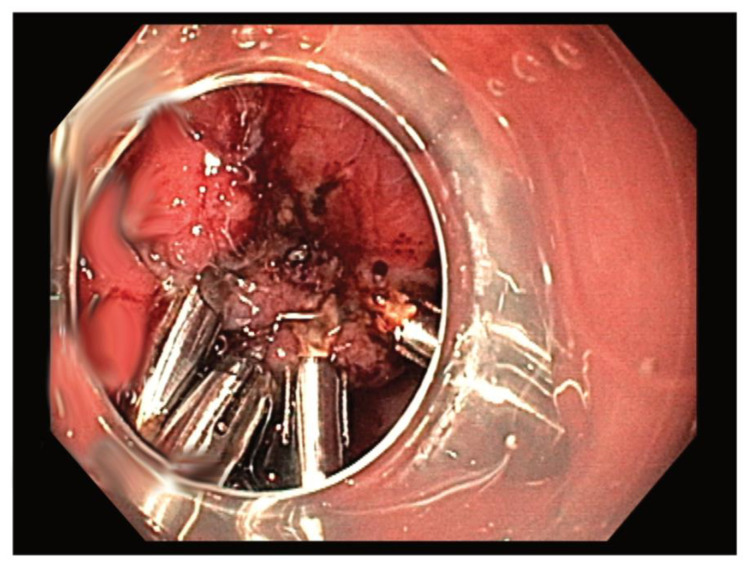
Endoscopic submucosal dissection of gastric GIST with closure of the defect using hemoclips.

**Figure 3 medicina-57-00625-f003:**
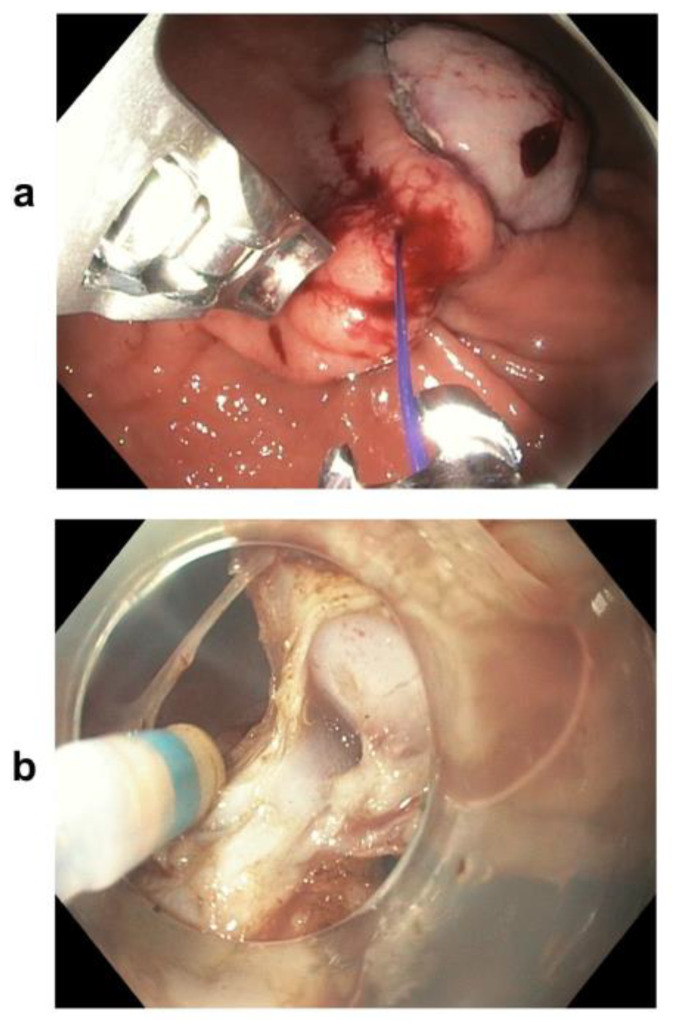
(**a**) Hybrid resection technique—gastric wall duplication (i.e., doubling the layers of the MP and serosa) through continuous suture below the tumour; (**b**) hybrid resection technique—the muscle layer was dissected between the lesion and the sutures.

**Figure 4 medicina-57-00625-f004:**
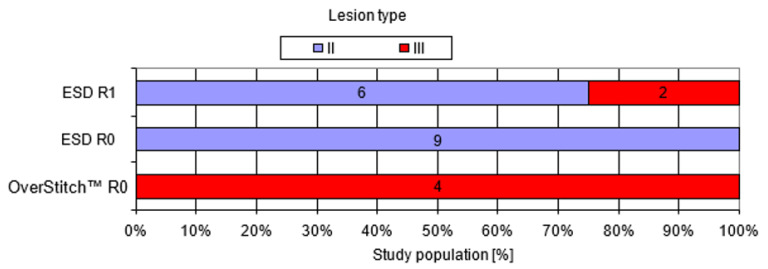
Discrepancies between lesion type and procedure types with (R0) and without (R1) therapeutic success. Abbreviations: ESD, endoscopic submucosal dissection.

**Figure 5 medicina-57-00625-f005:**
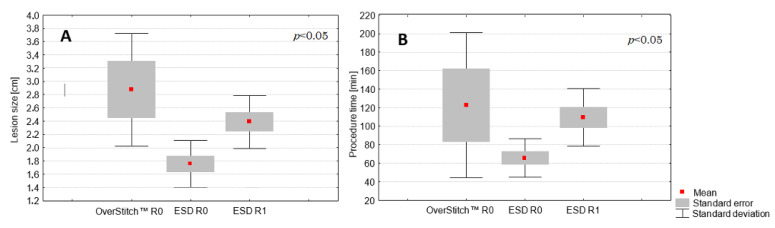
Comparison of the mean lesion size (**A**) and mean procedure time (**B**) according to procedure type with (R0) and without (R1) therapeutic success. The *p*-value was determined using ANOVA’s test. Abbreviations: ESD, endoscopic submucosal dissection.

**Figure 6 medicina-57-00625-f006:**
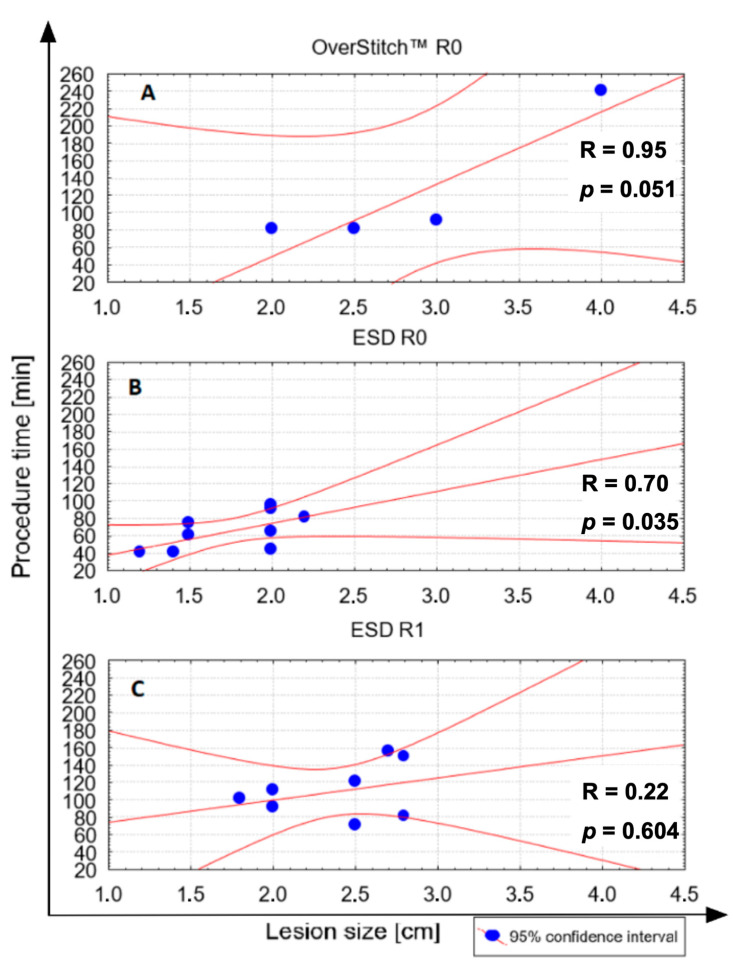
Assessment of the relationship between procedure time and lesion size in terms of selected technique—ESD vs. hybrid technique. (**A**) Hybrid OverStitch procedure with therapeutic success (R0); (**B**) ESD procedure with therapeutic success (R0); (**C**) ESD procedure without therapeutic success (R1). The Spearman’s rank (R) coefficient was calculated to analyse the association between procedure type, lesion size, and procedure time. Abbreviations: ESD, endoscopic submucosal dissection.

**Table 1 medicina-57-00625-t001:** Baseline characteristics.

	Overall (*n* = 21)	ESD (*n* = 17)	OverStitch (*n* = 4)	*p*-Value
Demographics				
Age (mean; SD)	70 (8)	70.5 (8.8)	68 (6.5)	>0.20
Female (*n*; %)	10 (48%)	9 (53%)	1 (25%)	0.58
Lesion type (*n*; %)				
II	15 (71%)	15 (88%)	0 (0%)	<0.005 *
III	6 (29%)	2 (12%)	4 (100%)	
Lesion localisation (*n*; %)				
Body	12 (57%)	11 (65%)	1 (25%)	<0.01 *
Antrum	7 (33%)	6 (35%)	1 (25%)	
Fundus	2 (10%)	0 (0%)	2 (50%)	
Procedural aspects				
R0 resection (*n*; %)	13 (62%)	9 (53%)	4 (100%)	0.13
Procedure time (min) (mean; SD)	93.1 (45.35)	86.2 (33.9)	122.5 (78.5)	0.15
Lesion size (cm) (mean; SD)	2.21 (0.64)	2.05 (0.49)	2.88 (0.85)	0.016 *
Path result	Confirmed GIST < 5 mitoses/50 HPF (very low risk)			
Adverse events rate (%)	0			

Continuous variables are expressed as the mean (SD) unless otherwise noted as *n* (%). The *p*-value was determined by comparing the procedures (OverStitch and ESD) using an independent sample Student’s *t*-test, Fisher’s exact test, or Pearson’s χ^2^, as appropriate. * Boldface type indicates a significant *p*-value (*p* < 0.05). Abbreviations: ESD, endoscopic submucosal dissection; GIST, gastro-intestinal stromal tumour; HPF, high power field; SD, standard deviation.

## Data Availability

Not applicable.

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
