# Peer review of "On the Track of New Endoscopic Alternatives for the Treatment of Selected Gastric GISTs—A Pilot Study"

_medicina, 2021, doi:10.3390/medicina57060625_

Round 1
Reviewer 1 Report
The authors demonstrated that the hybrid technique combining endoscopic resection and endoluminal suturing might be an alternative for gastric GISTs with large size and high MP connection grades (Type >I), with advantages on ESD although the study design was retrospective and the sample size was small. This paper is an interesting work and well written. There was no major comment. Minor comments were as follows:
- There are some mistakes in statistical analysis. The authors stated that categorical variables were assessed by using Fisher’s exact test and Pearson’s chi square test. In Table 1 and Table 2, the number in the OverStitch group was four; therefore, Fischer’s exact test should be applied to evaluate the differences between the ESD group and the OverStich group.
- The average procedure time of ESD was significantly shorter than that of OverStich86.2±9 min. However, in Table 1, p value was 0.15. Which is right?
- EFTR should be spelled out at the first appearance.
- Reference numbers were not match to those in text. At least, reference number 11, 12, 13, 14 and 15 were wrong.
Author Response
Dear Reviewer,
We greatly appreciate your interest in this study and value the constructive comments, which have, in our view, improved the manuscript. We have clarified all concerns and corrected the manuscript according to all suggestions. The reference list was correctly labeled, and all changes in the manuscript are indicated the “track changes” function in Microsoft Word.
Please find a detailed point-by-point response to the reviewers below.
- There are some mistakes in statistical analysis. The authors stated that categorical variables were assessed by using Fisher’s exact test and Pearson’s chi square test. In Table 1 and Table 2, the number in the OverStitch group was four; therefore, Fischer’s exact test should be applied to evaluate the differences between the ESD group and the OverStich group.
Thank you for this valid point. The p value was corrected in Table 2 as appropriate using mostly Fischer’s exact test for categorical variables. The Pearson’s chi square test was used to assess the differences between lesion localization in both groups as there are three variables, and the Fischer’s exact test is impossible to perform.
- The average procedure time of ESD was significantly shorter than that of OverStich86.2±9 min. However, in Table 1, p value was 0.15. Which is right?
This point is well taken, thank you. The statement was corrected in section Results.
- EFTR should be spelled out at the first appearance.
Thank you for your comment. EFTR abbreviation was explained.
- Reference numbers were not match to those in text. At least, reference number 11, 12, 13, 14 and 15 were wrong.
All references have been improved.
Also, we correctly defined the inclusion criteria, which were wrongly spelled, but the meaning was the same and it didn't impact the results.

Reviewer 2 Report
Thank you for submitting your paper.
Raiter et al. retrospectively investigated ESD or HT for gastric GISTs. Although this manuscript is well written, following aspects are missing that might improve the value of this article.
Major comments
Results, Table2
The cost and postoperative recovery period can be discussed by explaining the time to discharge after ESD and HT, so consider explaining period of hospital stay or time to discharge after the procedures in ‘Results’ and ‘Table 2’.
P3 L74 Inclusion criteria
The authors described a GIST size of 20-40 mm as ‘inclusion criteria’ for this study, but some lesions in the ESD group appear to be 20 mm or less, as estimated from the description in Table 2. Could you comment on this?
P4 L96
Please explain the definition of R0 resection. R0 resection is usually considered to be free of residual tumor cells on both the horizontal and vertical margin by pathological evaluation of the resected specimen. Also, describe the horizontal, vertical margin and tumor depth in postoperative pathological results of ESD and HT.
P8 L137, Fig6
The meaning of Fig6 is difficult to understand, so please explain in detail.
P10 L176
The authors are discussing the costs of ESD and HT, but there is no specific cost explanation, so please discribe them.
Minor comments
P2 L54-56
Cite the appropriate references that correspond to the content of the text below. ‘Type II and III GISTs …during endoscopic resection (Figure1).’
P3 L88, Fig2
The authors described the procedure in Fig2 as "clip closure of the defect," while Fig2 is a photo of a dual-knife submucosal resection. Please change the quoted part in ‘Figure 2’ or change the photo.
Table1, RESULTS-Characteristics of the study population
The author described the patient backgrounds with suspected pre-existing diseases in Table 1, but these are not mentioned in the main text. Delete it from the ‘Table1’ if you don't need it, or describe it in the main text if you need.
P8 L138
I think ‘EFTR’ is the first appearance, so please write all the spellings.
P9 L162
‘lower’ should be changed to ‘smaller’.
P15 Fig1
Please describe the explanation of the figure (meaning of each layer etc.). Describe each type in a legend and provide a ref to Fig1.
Ref
Since the reference notation in the text is different, please unify the format.
Author Response
Dear Reviewer,
We greatly appreciate your interest in this study and value the constructive comments, which have, in our view, improved the manuscript. We have clarified all concerns and corrected the manuscript according to all suggestions. The reference list was correctly labeled, and all changes in the manuscript are indicated the “track changes” function in Microsoft Word.
Major comments
Results, Table2
The cost and postoperative recovery period can be discussed by explaining the time to discharge after ESD and HT, so consider explaining period of hospital stay or time to discharge after the procedures in ‘Results’ and ‘Table 2’.
We thank the reviewer for this comment. The cost of hybrid technique (HT) is nine times higher than the standard ESD, but with HT we are able to remove GISTs more advanced than we could with ESD (usually lesions of this size must be resected surgically). The length of the postoperative stay for both resection techniques is short, patients after ESD were discharged on the first postoperative day (for type II) or on the second postoperative day (for type III) while patients after HT were discharged (on average) on the third postoperative day, earlier than after standard surgical treatment. This information was added to the text.
P3 L74 Inclusion criteria
The authors described a GIST size of 20-40 mm as ‘inclusion criteria’ for this study, but some lesions in the ESD group appear to be 20 mm or less, as estimated from the description in Table 2. Could you comment on this?
Thank you, this issue indeed requires explanation. The criteria were wrongly spelled, by written mistake. We included all lesions type II and III, and all resected with ESD and HT. The tumors resected were 12 to 40 mm in size. That was improved and cleaned. However, did not affect the analysis of the results, and did not impact the study.
P4 L96
Please explain the definition of R0 resection. R0 resection is usually considered to be free of residual tumor cells on both the horizontal and vertical margin by pathological evaluation of the resected specimen. Also, describe the horizontal, vertical margin, and tumor depth in postoperative pathological results of ESD and HT.
We appreciate the reviewer's comment, and according to this, the R0 definition was added. Regretfully, we cannot precisely describe the horizontal, vertical, or depth margin, as in our center, pathologists conclude only that “all margins of resected lesion are free”.
P8 L137, Fig6
The meaning of Fig6 is difficult to understand, so please explain in detail.
Thank you for this suggestion, we added some explanation about Figure 6 in section Results.
P10 L176
The authors are discussing the costs of ESD and HT, but there is no specific cost explanation, so please describe them.
We thank the reviewer for this comment. The cost of hybrid technique (HT) is nine times higher than the standard ESD, but with HT we are able to remove GISTs more advanced than we could with ESD (usually lesions of this size must be resected surgically). The length of the postoperative stay for both resection techniques is short, patients after ESD were discharged on the first postoperative day (for type II) or on the second postoperative day (for type III) while patients after HT were discharged (on average) on the third postoperative day, earlier than after standard surgical treatment. This information was added to the text.
Minor comments
P2 L54-56
Cite the appropriate references that correspond to the content of the text below. ‘Type II and III GISTs …during endoscopic resection (Figure1).’
Thank you for raising this point, the appropriate reference has been added.
P3 L88, Fig2
The authors described the procedure in Fig2 as "clip closure of the defect," while Fig2 is a photo of a dual-knife submucosal resection. Please change the quoted part in ‘Figure 2’ or change the photo.
Thank you for highlighting this. The correct figure was added.
Table1, RESULTS-Characteristics of the study population
The author described the patient backgrounds with suspected pre-existing diseases in Table 1, but these are not mentioned in the main text. Delete it from the ‘Table1’ if you don't need it, or describe it in the main text if you need.
We are grateful for that comment. The data regarding diseases has been removed from the Table 1. Furthermore, tables 1 and 2 have been merged to improve readability. All references in text to table 2 have been removed.
P8 L138
I think ‘EFTR’ is the first appearance, so please write all the spellings.
We thank you for highlighting this issue, we expanded the abbreviation of EFTR.
P9 L162
‘lower’ should be changed to ‘smaller’.
We appreciate the meticulous correction, the “lower” has been changed to “smaller”.
P15 Fig1
Please describe the explanation of the figure (meaning of each layer etc.). Describe each type in a legend and provide a ref to Fig1.
Thank you for your valuable comment. Figure 1 has been improved according to your suggestions.
Ref
Since the reference notation in the text is different, please unify the format.
Thank you for that comment. All references have been unified and correctly fitted to the main text.
Round 2
Reviewer 2 Report
Thank you for the revision.
I think the current version will be sufficient for publication.
Author Response
Once again, we would like to express our gratitude for the highly valuable review with comments improving our manuscript. We truly appreciate all Reviewer' efforts.
Thank you,
Yours sincerely
